# Easily Processable, Highly Transparent and Conducting Thiol-Functionalized Reduced Graphene Oxides Langmuir-Blodgett Films

**DOI:** 10.3390/molecules26092686

**Published:** 2021-05-04

**Authors:** Ki-Wan Jeon

**Affiliations:** Department of Environmental Energy and Chemistry, Silla University, Busan 46958, Korea; kiwan@silla.ac.kr

**Keywords:** thiol-functionalized reduced graphene oxide, transparent mRGO thin film, Langmuir-Blodgett film, highly conducting reduced graphene oxide thin film.

## Abstract

We report synthesis and fabrication of highly thionated reduced graphene oxide and its Langmuir-Blodgett (LB) film without an LB trough. As the synthesized product, mercapto reduced graphene oxide (mRGO) contains high thiol content estimated from XPS, corresponding to a surface coverage of 1.3 SH/nm^2^. The mRGO LB film shows two electronic transport properties, following Efros-Shklovskii variable-range hopping (VRH) and Mott VRH at low and high temperature, respectively. Optical and band gap of the LB film was estimated from Tauc plot and semi-logarithmic-scale plot of sheet resistance versus temperature to be 0.6 and 0.1 eV, respectively. Additionally, the sheet resistance of the mRGO LB film depends on the quantity of the thiol functional group with the same transmittance at 550 nm (500 kΩ for mRGO, 1.3 MΩ for tRGO with 92% transmittance).

## 1. Introduction

Since graphene was successfully isolated in 2004, it has emerged as a fascinating material for many potential applications due to its extraordinary electronic properties [1,2,3]. However, bulk production of chemically synthesized graphene or mechanically exfoliated graphene has been a challenging task for the past few years due to its utilization in many potential areas. Chemical modification or functionalization of the surface has been used as an easy pathway to tune various physical as well as chemical properties of carbon nanomaterials over the years [4,5,6]. In this regard, chemical modification of graphene has become a promising strategy to produce large quantities of graphene for different potential application purposes. Oxygen functionalized graphene is an electrically insulating material which is not desirable for many applications. To date, much research has paid attention to the manipulation of both the physical and chemical properties of graphene through chemical modification or reduction of oxygen functionalized graphene, namely graphene oxide (GO) [7,8].

To date, oxygen functional groups on chemically modified graphene generated during GO synthesis have been utilized for different purposes. Mainly, carboxyl and hydroxyl groups on graphene have been utilized as a linking unit [9,10]. The utilization of chemically modified graphene with multifunctional groups, however, could open up new directions of potential research. Among diverse functional groups, the thiol functional group is favorable for various purposes such as a cross-linking group through disulfide formation [11], click chemistry in various reaction media [12], self-assembly monolayers (SAMs) on gold surfaces [13], heavy metal scavenging [14], biosensors [15], biomedical applications [16], and passivation and stabilization of noble metal nanoparticles for biological applications [17].

Despite the existence of various thionation routes for organic compounds in general, their applications have been rarely reported for thionation of GOs [18]. One notable exception is a quite recent report by Thomas and her collaborators in which a thiol-functionalized GO was obtained through a nucleophilic reaction of thioacetate with an epoxide on GOs [19]. The resulting new material exhibits a significant amount of thiols (4 at%) with C:O:S ratios of 4.1:1:0.22. It is not clear if the product was electrically conducting as the overall content of the oxygen-functional groups was rather high. In our previous work, concomitant thionation and reduction of graphene oxides were carried out to achieve high density thiol functional groups on reduced graphene oxides (C:O:S = ~17:1:2) through solid-gas metathetical reactions employing gaseous boron sulfide (B_x_S_y_) molecules [20], but unfortunately the resulting products were rather strongly re-stacked due to the required pre-drying and the high reaction temperatures (over 500 °C). Thionation of GOs [21] and oxidized carbon nanotubes [22] have been reported with P_4_S_10_ as a thionating agent. While the amount of the thiol groups was low (0.6 at%) [22] or unreported [21], they could successfully anchor CdSe quantum dots and silver nanoparticles on the surface of the carbon nanomaterials. In a somewhat distant work, thiourea has been successfully used to metathetically replace hydroxyl groups on sp3-carbon on nanodiamond at a high yield of up to 85% [18]. Herein, we report an alternative thionation route and fabrication of its Langmuir-Blodgett (LB) film without using an LB trough.

## 2. Results and Discussion

The summary of chemical compositions and the existing functional groups in the products are given in Table 1. The sample names were designated according to the reaction temperatures (120 °C, 150 °C, and 180 °C). The thermally reduced graphene oxides (tRGOs) listed in Table 1 were prepared as a control by using the same reaction conditions except adding P_4_S_10_. Our synthesis employs P_4_S_10_ as the thionating agent that converts oxygen-functional groups on graphene oxides (GOs) into sulfur-functional groups through metathesis. Although P_4_S_10_ had been utilized for the same purpose in previous reports, the amounts of the thiols in the products were rather small based on the XPS data. In principle, P_4_S_10_ acts as a highly efficient thionating agent for various organic compounds including alcohols, ketones, ethers, and esters, with a reasonably high reaction yield (50%–98%). The selectivity for some reactions can be less impressive [23], but this is in fact advantageous in our case where various different oxygen-functional groups on GOs may be able to be converted into their sulfur-analogues in a single step under the same reaction condition. Typical GOs contain a significant amount of oxygen-functional groups with C/O ratio of 2 to 4 and most of the oxygen atoms exist in the form of carbonyl, hydroxyls/ether, and carboxyl groups (~6:~2:1) [24]. It would be desirable to thionate or potentially remove all these oxygen-functional groups simultaneously in a single reaction step.

Unlike the previous report where a refluxing condition was employed with dimethyl formamide (DMF) as a solvent [21], we employed a solvothermal reaction condition with pyridine. P_4_S_10_ is more effective when used with pyridine as a solvent or co-reagent, as reported previously [23], by reacting readily with pyridine to form a zwitterionic compound (P_2_S_5_·2C_5_H_5_N) which does not decompose easily and remains effective even above 170 °C. This high thermal stability would work favorably in our solvothermal condition at temperatures from 120 °C to 180 °C, much higher than the boiling point of pyridine (115 °C). It is noted that an appreciable amount of water (~0.1%) in pyridine solvent such as commercial pyridine is necessary to achieve the reported good yields, as exemplified with thionation of ketones [25]. Thionation of alcohols by 1 needs a closer look in the context of our work. Its reaction with alkylalcohols typically produces dialkyldithiophosphoric acid instead of thiols [26], unlike phenolic hydroxyl groups [27]. Since the carbon atoms in GO have an aromatic nature, it is expected that the hydroxyl groups in GO would be thionated to the corresponding thiols. Figure 1 shows the schematic reaction of alcohol and ketone using zwitterionic compound (P_2_S_5_·2C_5_H_5_N).

The elemental analysis and the identification and quantification of the functional groups of the initial (GOs) and the final products (mRGOs and tRGOs) were carried out by employing XPS and the results were summarized in Table 1. The XPS spectra of the mRGO-180 are shown in Figure 2, as a representative example. The spectra for the rest of the samples are shown in Appendix A. The relative atomic ratios of C, O, S, and P were obtained from the survey scans and summarized in Table 1. The C:(O+S) ratios for mRGOs indicate the overall number of carbon atoms relative to the combined number of oxygen- and sulfur-functional groups. The ratio increases only slightly from 5.8 to 5.9 as the reaction temperature increases from 120 °C to 180 °C. The temperature effect is more pronounced in the relative amounts between oxygen- and sulfur-functional groups. In mRGO-120, the amount of oxygen atoms is larger than that of sulfur at the ratio of 1:0.7. However, the sulfur amount is larger than the oxygen amount by the factor of 1.2 in mRGO-180. The decrease in the amount of the oxygen-functional groups was observed also for the tRGO series at a similar rate, as found in Table 1, even though the products were obtained without P_4_S_10_. This indicates that the reduction of the oxygen-functional groups is mainly a temperature effect. C/(O+S) ratios in Table 1 show that the relative number of carbon atoms with respect to the total number of oxygen and sulfur atoms are within the typical range found for RGOs from various methods. Hydrothermally reduced GOs in water at 180 °C were reported to have a C/O ratio of 5.3. The C/O ratios from 2.8 to 4.4 were obtained for the RGOs prepared under a refluxing condition in various organic solvents (boiling point: 153 °C to 204 °C) [28]. Judging from the C/O and C/(O+S) ratios, the amounts of the functional groups in both mRGOs and tRGOs are slightly larger than these numbers but less than the amounts reported for the RGOs prepared from chemical reduction. For example, hydrazine routes lead to the RGOs with C/(O+N) ratios from 7 to about 8 [29,30].

The nature of the carbon atoms was examined by deconvoluting the high-resolution XPS spectra in the C1s, O1s, and S2p energy regions (Table 1). Figure 2b–d show the deconvolution of the peaks for mRGO-180, as an example. The XPS spectra of the other samples including tRGO-180 and details of the deconvoluted peaks are given in Appendix A. At least 80 at% of the carbon atoms are graphitic for all the mRGO and tRGO samples based on the deconvoluted C1s peaks. The number increases only slightly with increasing temperature. About 6 to 9 at% of the carbon atoms are functionalized in all those samples, while in the mRGOs about 6 to 8 at% of the carbon atoms have a thiol group. The amount of the thiol groups increases as the reaction temperature increases; so too does the amount of thiocarbonyl groups. The maximum sulfur content (7.6 at%) found in the mRGOs is more than 10 times higher than the highest amount reported for nanostructured carbon materials with P_4_S_10_ [22].

Assuming that there are no carbon defects in the structure and that the thiols are equally distributed on both sides of the graphene sheets, the thiol content estimated from the XPS data corresponds to one SH in the area of 6.5 unit cells on average, or equivalently a surface coverage of 1.3 SH/nm^2^ on each side of a graphene sheet. The presence of the thiol groups can be confirmed by comparing the ATR FT-IR spectrum of mRGO-180 with that of tRGO-180 (Appendix A). While the tRGO-180 shows no distinct peak in the spectrum range, the mRGO-180 clearly exhibits an absorption peak centered at 665 cm^−1^ which has been attributed to C‒S stretching band in thiols in the literature [31].

Figure 3a,b show dispersions of the mRGO-180 and tRGO-180, respectively, in deionized water (left), ethanol (middle), and N,N-dimethylformamide (DMF) (right), prepared by ultrasonication for 15 min. Both mRGOs and tRGOs show a good dispersibility in all those solvents upon visual inspection. After centrifugation at 4000 rpm for 10 min, however, most of the tRGOs precipitated out in all solvents (Figure 3d) while mRGOs remained in both ethanol and DMF (Figure 3c). Interestingly, the mRGOs exhibits a relatively good dispersibility in water even after the centrifugation, as can be seen in Figure 3c. The good dispersibility of the mRGOs in both water and organic solvents is rather remarkable in contrast to most of GOs, RGOs, and their derivatives that show rather exclusive dispersibility in either water or organic solvents [6,32]. Since both mRGO-180 and tRGO-180 have similar amounts of functional groups, the main reason for their contrasting dispersibility behaviors may be rooted in the fact that about half of the functional groups are thiols in the case of mRGO-180. Aromatic thiols in water are more acidic (pKa ~ 6) than aromatic alcohols in water (pKa ~ 10) [33]. Deprotonation of thiols would provide negative charges to mRGOs so that mRGOs can disperse well in water.

To investigate morphology of the mRGO sheets, the mRGO-180 shown in Appendix A was prepared by freeze-drying its aqueous dispersion. The fluffy agglomerates in the images consist of severely wrinkled and folded sheets, indicating that the restacking of the graphene sheets was not significant in the product or during the freeze-drying process. This corroborates our observation that mRGOs could be dispersed well in water (Figure 3). The individual mRGO sheets were more closely examined by using a scanning transmission electron microscope equipped with energy dispersive X-ray spectroscopy (STEM-EDS) which is shown in Figure 4. The sulfur elements were well distributed over the entire mRGO sheet and the mRGOs were well exfoliated based on the highly transparent sheet. Additionally, the average O:S atomic ratios of the mRGO were 1:0.9(2) estimated from STEM-EDS (Appendix A), which is in agreement with the XPS results. The amount of carbon was not quantified because of interference by the presence of the carbon grid.

The reduced nature of the mRGOs could be verified from UV-Vis spectra of the samples, as shown in Appendix A. The π→π* transition occurs at 273, 260 and 229 nm for mRGO-180, tRGO-180 and GO, respectively, in their spectra. The largest red shift of the transition peak (i.e., the smallest energy gap between π and π* bands) for mRGO-180 may indicate the highest degree of π conjugation in the graphene sheets among the three samples, as often explained in the literature [32,34]. However, mRGO-180 and tRGO-180 have similar amounts of functional groups, and thus the largest red shift for mRGO-180 cannot be solely due to the π conjugation effect. In fact, the ID/IG ratio in the Raman spectrum (Appendix A) is the smallest (0.91) for tRGO-180 while GO and mRGO-180 have similar values (1.08 and 1.12, respectively). In other words, as far as phonon motions (atomic vibrations) within the graphene sheets are concerned, the graphitic domains are almost the same in size for GO and mRGO-180. The additional effect for the smaller π-π* energy gap might be from the fact that the diffuse 3s and 3p orbitals of sulfur interact more effectively with π-orbitals in the graphene sheet than the compact 2s and 2p orbitals of highly electronegative oxygen atoms, leading to larger energy band widths and a smaller band gap, as found in organic superconductors, for example.

The small energy gap between π and π* bands in mRGO-180 is evident also from its Tauc energy gap. Appendix A shows Tauc plots of (A/λ)^1/2^ versus *hν* for GO and mRGO dispersions obtained from their UV-Vis absorption spectra where A is absorbance at the wavelength λ. By extrapolating the linear region of the curves to the energy axis, the Tauc gaps of GO, mRGO-120, mRGO-150, and mRGO-180 are estimated to be 3.07, 1.38, 1.16, and 0.63 eV, respectively. The Tauc gap estimated for mRGO-180 is smaller than the one (0.85 eV) reported for the highly reduced graphene oxides through a chemical process using NaBH_4_, [35] although the former has a lower degree of reduction than the latter. The Tauc gap for the corresponding tRGO-180 was not obtained due to its poor dispersibility in water, whereas the tRGO-150 sample showed a Tauc gap of 1.30 eV, a value much larger than that for mRGO-180.

In order to examine the electrical properties of mRGOs, the mRGO-180 Langmuir-Blodgett (LB) film was fabricated by transferring mRGO Langmuir film prepared without using a Langmuir-Blodgett trough in a self-assembled manner by dripping mRGO dispersion in ethanol onto the water surface. The SEM of the mRGO LB film in Figure 5 indicates that the graphene sheets are physically in contact, which could result in low sheet resistance. The thickness of the individual mRGO sheets was about 1.4 nm from AFM studies of the LB films (Appendix A), suggesting complete exfoliation of the mRGO in its dispersion in ethanol. The optical property of mRGO could be similar to that of graphene or RGOs according to the estimated absorption coefficient of mRGO (see Appendix A) [36]. From the transmittance (Figure 6a) of the mRGO-180 LB film at 550 nm wavelength, [37] however, the average number of the graphene sheets was estimated to be ~4, which can be due to the severe wrinkles that were clearly noted in both the SEM and AFM images. The sheet resistance (R_□_) (Figure 6b) of the mRGO-180 LB film on a slide glass was ~450 kΩ/sq at room temperature which is much lower than that of tRGO-180 (16 MΩ/sq, Appendix A). It is more interesting that the sheet resistance of the mRGO LB film could be controlled with the amount of sulfur functional groups. These values are still higher than what have been reported for highly reduced graphene oxides, but it is noted that our LB films were prepared without using an LB trough [2,38].

Chemical functionalization of graphene creates atomic disorders and thus has a significant effect on the electron transport properties of the material. In order to elucidate the electron conduction mechanism for mRGOs, the relationship between R_□_ and temperature (T) was studied by using three different mechanisms: conventional hopping, Mott variable range hopping (Mott-VRH), and Efros-Shklovskii variable range hopping (ES-VRH) models [39,40]. While the hopping model is described by an Arrhenius behavior of R_□_, the VRH can be in general characterized in the Ohmic regime with a low bias voltage as
$$R_{□}(T)=R_{0}{\left(\frac{T_{0}}{T}\right)}^{p},$$
where R_0_ is a prefactor, *T*_0_ is a characteristic temperature, and *p* is a characteristic exponent which is 1/3 for Mott-VRH [39] and 1/2 for ES-VRH [40]. Appendix A shows the semi-logarithmic-scale plots of R_□_ vs. T^–1^ (conventional hopping), T^–1/2^ (ES-VRH), and T^–1/3^ (Mott-VRH) with extrapolating solid lines in both high temperature and low temperature regions. By comparing the linear extrapolating lines, it can be concluded that the electron transport in the sample follows the ES-VRH mechanism, while at high temperature it shows a Mott-VRH behavior. It has been recognized in the literature that such a crossover from ES- to Mott-VRH with increasing temperature takes place when the material has an intermediate degree of disorders [41]. In the ES-VRH region, it is possible to estimate the size of the so-called localization length (ξ) from the slope of the extrapolating line [41]:$$T_{0}=\frac{2.8e^{2}}{4\pi \varepsilon \varepsilon_{0}k_{B}\xi },$$
where *e*, *ε*, *ε*_0_, and *k_B_* are the electron charge, the dielectric constant of the material, the vacuum permittivity, and the Boltzmann constant, respectively. The localization length is further related to the electronic band gap (Eg) through the following approximate relationship [41]:$$E_{g}\sim \frac{hv_{F}}{2\pi \xi },$$
where *h* and *v_F_* are the Plank constant and the Fermi velocity of the material [41]. By employing the values for the parameters from the literature, the ξ value for the mRGO-180 was estimated to be 6 nm and the corresponding *E_g_* was 0.1 eV. These values are comparable or indicative of an “innately” high conducting nature of the material in comparison to the chemically reduced graphene oxides with a similar degree of reduction (the carbon sp2 content = 80%; ξ = 3 nm; E_g_ = 0.2 eV) [41]. The observed relatively large sheet resistance for the mRGO-180 might be due to an ineffective physical contact among the graphene sheets in the LB films.

## 3. Materials and Methods

### 3.1. Thionation of GOs

GOs were prepared by using modified Hummers method through oxidation of natural graphite flakes with strong oxidants. The detailed procedure is presented in our previous report [42]. The solvent exchange of the GOs dispersion in water was performed with the following: 50 mL of the purified GOs dispersion in water (2 mg/mL) was added to 250 mL round-bottom flask and the flask was attached to rotary evaporator. Water in the flask was evaporated under reduced pressure and gentle heating (60 °C) by a rotary evaporator until GOs becomes sludge. Subsequently, 50 mL of pyridine was added to the flask and sonicated for 20 min and the rotary evaporation was continued until about 10 vol% of the liquid remained, which was repeated one more time. The resulting sludge in pyridine was sonicated for 20 min and a certain amount of P_4_S_10_ (Sigma-Aldrich, St. Louis, MO, USA, 99%) was added into the sludge and dissolved. In our preliminary experiments, it was found that an excess amount of P_4_S_10_ disfavored the thionation under our reaction condition and thus all the subsequent reactions were carried out with a stoichiometric amount of P_4_S_10_ with respect to the total amount of the oxygen-functional groups. In a typical solvothermal reaction, 555 mg of P_4_S_10_ (stoichiometric amount of P_4_S_10_) were slowly added to 100 mg of GOs in 38 mL of pyridine in a Teflon-lined autoclave of capacity of 45 mL. It was then placed in a laboratory oven preset at 120 °C, 150 °C, or 180 °C for 15 hrs. The solid product was collected via vacuum filtration and it was washed several times with deionized water and ethanol to remove all the unreacted starting materials and by-products. The mRGOs in water were freeze-dried for further characterization. This final product could be easily dispersed in various solvents including H_2_O, dimethyl formamide (DMF), dimethyl sulfoxide (DMSO), and N-methyl pyrridone (NMP) by sonication for 15 min to achieve the stable mRGO dispersions in a wide range of solvent media. For comparison, tRGOs were synthesized through the same reaction condition as that of as-synthesized mRGOs without P_4_S_10_.

### 3.2. Fabrication of mRGO Thin Films on Glass Substrate

Langmuir-Blodgett (LB) films of the mRGO were fabricated without using a Langmuir-Blodgett trough. A dispersion of mRGO in ethanol (6 μg/mL) was prepared first by mixing 10 mg of a wet mRGO sample (~0.6% solid weight in ethanol) with 10 mL of absolute ethanol and subsequently sonicating the mixture in an ultrasonication bath for 15 min. The dispersion was then centrifuged at 4000 rpm for 10 min and the gray supernatant was collected. By using a squeeze pipet (5 mL), the supernatant dispersion was dripped gently into a 6-cm diameter petri dish half full of deionized water. After adding about six drops of the dispersion, thin gray patches (~ 1 mm) of mRGOs appeared on the surface of water. The small patches continuously drifted around on the surface until they were assembled into a large continuous film (mRGO Langmuir film) in a spontaneous manner.

The slide glass was cleaned by sonication in absolute ethanol and acetone for 15 min and subsequently rinsed with deionized water. The cleaned glass was dried by N_2_-gas stream. The mRGO Langmuir film was transferred on the glass surface by carefully scooping the Langmuir film up with the glass substrate. The transferred film on glass was dried by N_2_-gas stream and it was visually checked that the area of the glass surface was covered by a translucent gray film. The tRGO LB film can be fabricated with the same manner described above, but the similar quality (thickness and roughness) of the tRGO LB film could not be achieved due to poor dispersibility of the tRGO in ethanol.

### 3.3. Materials Characterization

X-ray photoelectron spectroscopic (XPS) measurements were carried out using a VG-220IXL spectrometer with a monochromated Al Kα radiation (1486.6 eV, line width 0.8 eV). The pressure in the analyzing chamber was kept at the level of 10^−9^ torr while recording the spectra. The spectrometer has the energy resolution of 0.4 eV. All the binding energies were corrected with reference to C(1s) at 284.6 eV. Deconvolution of the spectrum was done using the CASA software with the accuracy of 0.2 eV. Shirley background was used for the deconvolution. For the high-resolution C1s XPS spectrum was deconvoluted into the following three components: C–C (sp2- and sp3-hybridized peaks at 284.7 and 285.4 eV, respectively) [24], C–S (285.3 eV), [28] and C–O (hydroxyl or ether peaks at 286.5 eV) [43]. Carbon atoms with the C–S bond were not separately treated because the C1s binding energy of C–S (285.3 eV) is too close to that of C–C (sp3-hybridization) and thus could not be resolved, given the resolution of the XPS instrument (0.4 eV). The high-resolution O1s XPS spectrum was presented with the following oxygen functional groups: P=O (531.7 eV), [44] and/or oxygen binding energy in sulfonic functional group (531.2~532 eV), [45] as well as C-OH (532.7 eV) [28]. The high-resolution S2p spectrum was deconvoluted with three functional groups: C=S (S2p3/2 at 162.0 eV with FWHM of 1.4 eV; S2p1/2 at 163.2 eV with FWHM of 1.4 eV), C–SH (S2p3/2 at 164.0 eV with FWHM of 1.2 eV; S2p1/2 at 165.2 eV with FWHM of 1.2 eV), [46] and C-SO_3_H (S2p3/2 at 167.5 eV with FWHM of 1.4 eV; S2p1/2 at 168.7 eV with FWHM of 1.4 eV) [47]. The area ratio and splitting energy difference between S2p3/2 and S2p1/2 spin-orbit doublet peaks were 2:1 and 1.2 eV, respectively.

A Nicolet 6700 FTIR spectrometer (Thermo Scientific Nicolet) was used to collect attenuated total reflectance (ATR) FTIR spectra of ethanol and supernatant solution of mRGOs dispersed in ethanol. Scanning electron microscopy (SEM) studies were performed on mRGO LB films on Si wafer using an FEI XL-30 Environmental SEM using 10 keV electrons. Scanning transmission electron microscopy (STEM) images and elemental mapping images were acquired using JEOL 2010F (200kV) TEM/STEM equipped with a Schottky type field emission gun and EDAX thin window X-ray energy dispersive spectrometer (EDS) detector. For the STEM-EDS composition analysis, 8 different areas were examined and averaged out.

UV-Vis measurements were carried out using a Hewlett-Packard 8453 spectrophotometer using quartz cuvettes with a 1-cm path length. Surface topography images were obtained using atomic force microscope (Pico-Plus AFM, Molecular imaging, Agilent technologies). All AFM studies were performed in air using a tapping mode with SCANASYST-AIR tips (Bruker). The images were collected at a scan rate of 1.0 Hz in air.

The Raman spectra were collected using a custom-built Raman spectrometer in 180° geometry. The sample was excited using a 0.75 mW Compass 532 nm laser. The laser power was controlled using neutral density filters. The laser was focused onto the sample using a 50X superlong working-distance Mitutoyo objective with a numerical aperture of 0.42. The signal was discriminated from the laser excitation using a Kaiser laser band pass filter followed by a Semrock edge filter. The data were collected using an Acton 300i spectrograph and a back thinned Princeton Instruments liquid nitrogen cooled CCD detector.

After mRGOs LB films were fabricated onto 1 × 1 cm^2^ glass substrate, drops of sliver paint (2SPI Supplies/Structure Probe, Inc.) were applied at four corners of the sample to form dots of ~1.5 mm in diameter. After drying the silver paint ambiently, the sample was then loaded into a four-point probe device with four springs loaded pogo-pins pressed firmly onto the silver paint contact dots. The sheet resistance was then measured with the standard Van der Pauw scheme; a current (I) was sourced between the two contact dots along one side of the sample, and the voltage (V) measured across the contacts on the opposite side. The sample sheet resistance (R_□_) can be obtained as R_□_ = 4.53V/I. The current source (Keithley 6221) had an output impedance of ~1014 that was well suited for the measurement. The temperature was changed by slowly lowering the dipping probe into a dewar of liquid He, and temperature was measured using a calibrated silicon diode sensor. Both sample and diode sensor were in good thermal contact with the two faces of a thin block of copper.

## 4. Conclusions

Highly conducting reduced graphene oxides have been successfully prepared through development of a new thionation method. The maximum thiol content was higher than 7 at% which corresponds to one SH in the area of 6.5 unit cells on average, or equivalently a surface coverage of 1.3 SH/nm^2^ on each side of a graphene sheet. The new method employs P_4_S_10_ as an effective thionating agent in the pyridine solvent which acts as a co-reagent. The thionation was more effective at higher temperatures of up to 180 °C and thus allowed a concomitant thermal reduction of the graphene oxides under solvothermal conditions. Despite the similar degree of reduction, the thiol-functionalized reduced graphene oxides showed a much superior electrical conduction behavior than thermally reduced graphene oxides prepared without the presence of P_4_S_10_, while exhibiting a remarkable dispersibility in various solvents including DMF, alcohols, and even water. The estimated electronic band gap was only 0.1 eV, indicating an innately high conducting nature of the material in comparison to the chemically reduced graphene oxides with a similar degree of reduction. The unique features of high thiol functionality, high electrical conductivity, and good water dispersibility of the new material may allow its diverse applications including the self-assembly monolayers (SAM) on gold substrate, heavy metal scavengers and biosensors. Furthermore, it is envisaged that the new thionation method can be applied to other carbon nanomaterials, opening up new possible applications for those materials.

## Figures and Tables

**Figure 1 molecules-26-02686-f001:**
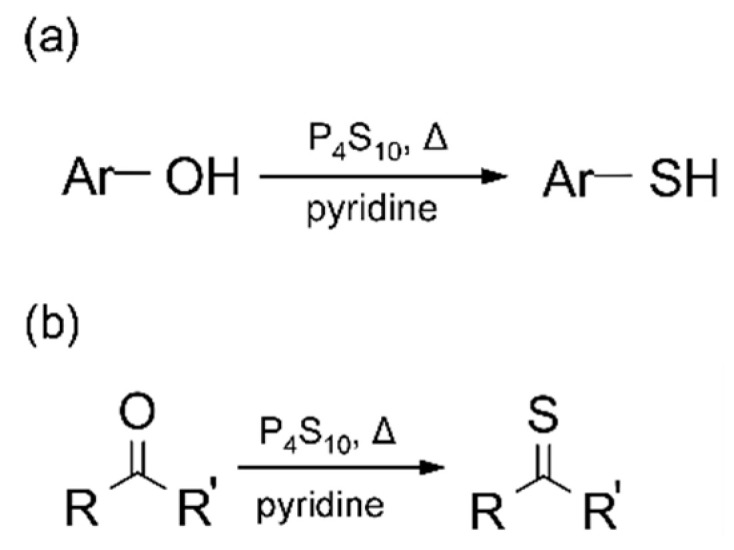
Schematic reaction of (**a**) alcohol and (**b**) ketone with Berzelius reagent in pyridine.

**Figure 2 molecules-26-02686-f002:**
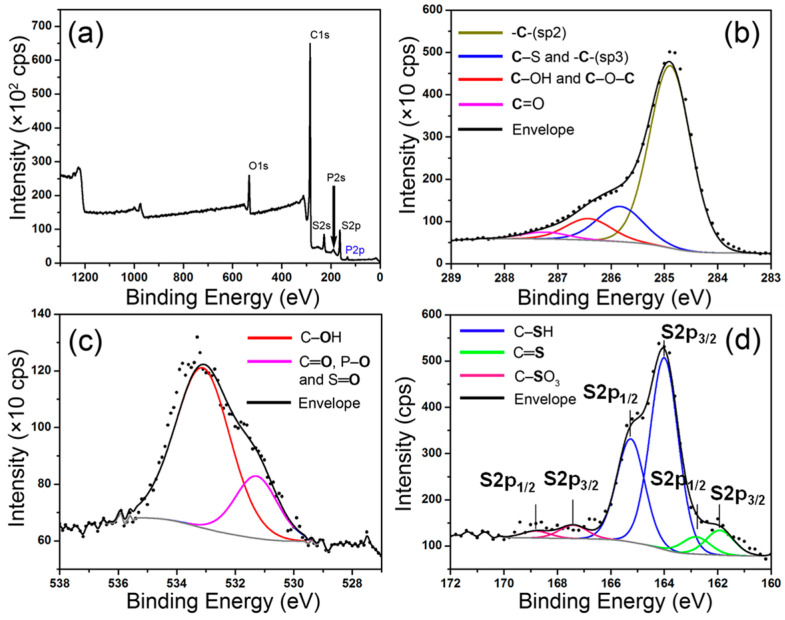
XPS spectra of mRGO-180: (**a**) wide scan; (**b**) high-resolution C1s; (**c**) high-resolution O1s and (**d**) high-resolution S2p.

**Figure 3 molecules-26-02686-f003:**
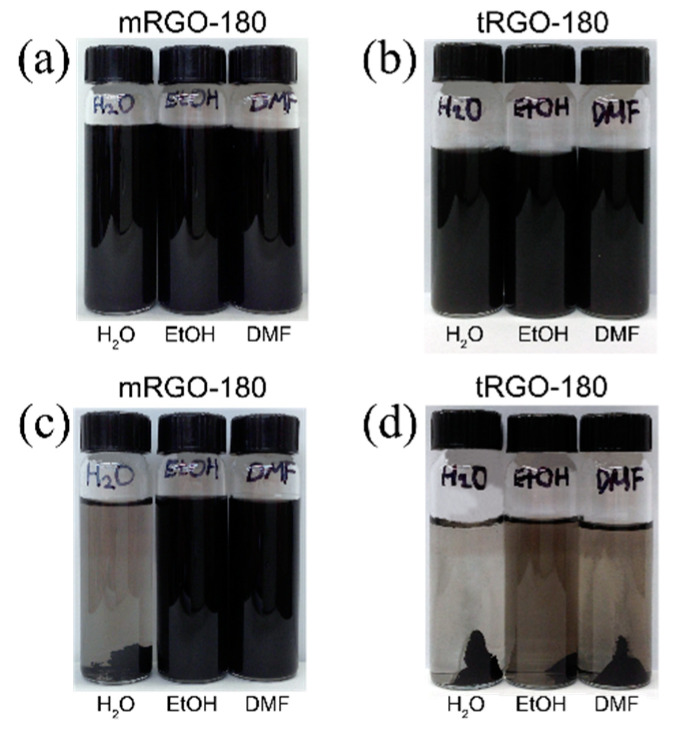
Photographs of (**a**) and (**c**) mRGOs and (**b**) and (**d**) tRGOs dispersion in water, ethanol (EtOH), and dimethylformamide (DMF) (**a**) and (**b**) right after sonication and (**c**) and (**d**) after centrifugation of (**a**) and (**b**) at 4000 rpm for 10 min.

**Figure 4 molecules-26-02686-f004:**
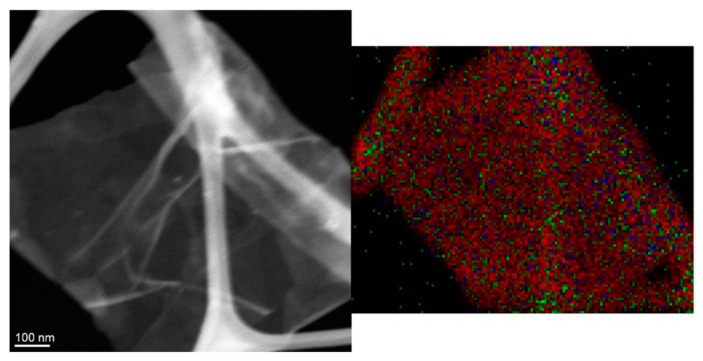
STEM-EDS elemental mapping image of mRGO, red, green, and blue dots indicate carbon, sulfur, and oxygen atoms, respectively.

**Figure 5 molecules-26-02686-f005:**
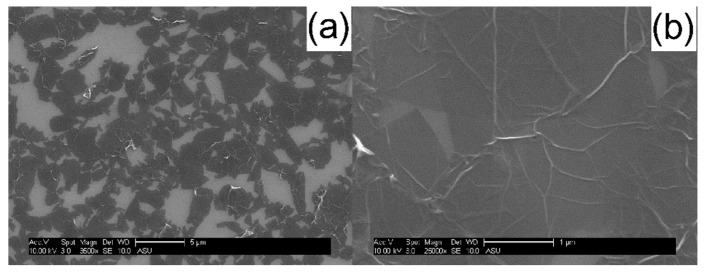
(**a**) Low magnification and (**b**) high magnification of SEM images of mRGO-180 LB film on Si wafer.

**Figure 6 molecules-26-02686-f006:**
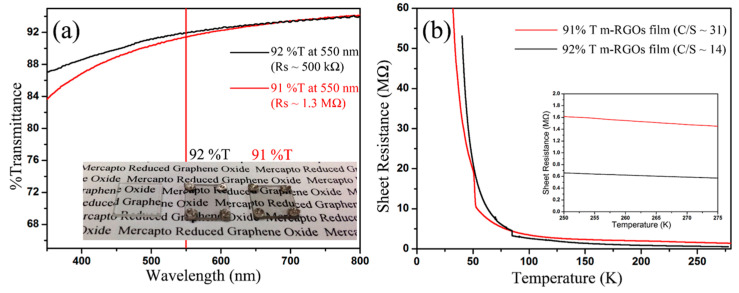
(**a**) Optical and (**b**) electrical properties of mRGO-180 LB film.

**Table 1 molecules-26-02686-t001:** Relative atomic ratios and the amounts of functional groups in GO, mRGO, and tRGO prepared at different reaction temperatures.

Sample	Relative Atomic Ratios			% C atoms Attached with Different Functional Groups						
	C:O:S:P	C/(O+S)	Graphitic ^a^	C-OH ^a^	C=O ^b^	COO ^a^	C-O-C ^b^	C-SH	C=S	C-SO_3_^−^
GO	2.1:1:0.07:0	1.96	41.4	24.3	8.5	3.6	18.8	0	0	3.3
mRGO-120	9.7:1:0.68:0.12	5.77	82.7	7.8	2.5	0	0	6	0.6	0.3
mRGO-150	11:1:0.91:0.18	5.76	82.6	6.5	2.5	0	0	6.7	1.0	0.6
mRGO-180	13:1:1.2:0.8	5.90	83.1	5.9	1.8	0	0	7.7	1.1	0.5
tRGO-120	4.9:1:0:0	4.90	79.6	10.2	10.0	0	0	0	0	0
tRGO-150	5.8:1:0.03:0	5.63	82.2	9.6	7.6	0	0	0	0	0.5
tRGO-180	6.6:1:0.06:0	6.23	83.1	10.3	4.8	0	0	0	0	0.9

^a^ estimated from high-resolution C1s XPS spectrum. ^b^ estimated from high-resolution O1s XPS spectrum.

## Data Availability

The data presented in this study are available in the article.

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
