# Peer review of "Easily Processable, Highly Transparent and Conducting Thiol-Functionalized Reduced Graphene Oxides Langmuir-Blodgett Films"

_molecules, 2021, doi:10.3390/molecules26092686_

Round 1

Reviewer 1 Report

The authors report the synthesis and fabrication of highly thionated reduced graphene oxide and its comparison with thermally reduced graphene oxides (tRGOs). The authors claim that despite the similar degree of reduction, the thiol-functionalized reduced graphene oxides showed a better electrical conduction behavior than thermally reduced graphene oxides prepared without the presence of P4S10, as well as exhibiting a remarkable dispersibility in various solvents including DMF, alcohols and even water. Therefore, this new rGO could be used in several applications taking into account its properties.

Overall, I would recommend the manuscript for publication in Molecules after the following major revision:

Explain in more detail how the LangmuirBlodgett (LB) film were fabricated, using the vertical dipping method? Which was the transference surface pressure? If it was using a SA method, why did you use the term LB? Maybe is better to use the term SA, to avoid confusion to the readers.

Why are the words Langmuir-Blodgett used in the title if really a self-assembly method is used? In addition, please re-write correctly in the title the word “Blodget” if this is keeping.

The relative atomic ratios of C, O, S and P were obtained from the survey scans, why are not used the high resolution spectra? Which would be more accurate

In Figure 2, please increase the size of the legends, imposible to read

The authors assume that there are no carbon defects in the structure and that the thiols are equally distributed on both sides of graphene sheets. Why do they not used for example, atomic force microscopy, to corroborate the non-presence of defects?

Where are in the main text figure 4 and 5 and 6?

Figure S5 does not correspond to XPS results, Figure S6 not correspond to the UV-vis spectra in the main text, please chech the numbering of the figures 

How have been the electrical properties of mRGOs  determined?

Reviewer 2 Report

Overview: In the presented manuscript, the author reports the synthesis of a highly thionated reduced graphene oxide (mRGO), and its Langmuir-Blodgett film. The author proposes a novel synthetic approach, in wich the thionation and reduction of graphene oxide is performed by employing phosphorus decasulfide as thionating agent and solvothermal reaction conditions. Moreover, the author reports an alternative method for the fabrication of mRGO Langmuir-Blodgett films. The mRGO was chracterized by the use of different methods, giving a quite complete morphological and optical characterization of the system. Higher solvent-dispersibility and electrical conductivity were observed in mRGO with respect to thermally reduced graphene oxide, that are associated by the author to the presence of sulphur-based functional groups.

The paper presents an original work, overall is well written and could be of interest to the surface science community, as well as in the fields of the electronic devices and nanomaterials. In my opinion, the Molecules Journal is an appropriate place to publish because it can reach each of these target groups  Hence, I recommend this manuscript for publication, after the following minor revisions.

General Comments: The manuscript is overall well written, the papers aims to present the properties of highly thiol-functionalized reduced graphene oxides films fabricated with an alternative method. However the Results and Discussion section turns out to be fragmentary, and the usefulness of some sets of data is not always clear.  Maybe the material presented in the manuscript should be divided in separated publications. Moreover, it is quite difficult, at least to me, to follow the discussion about the results, with the text referring to figures not included in the main text, forcing the reader to jump continuosly to the supporting info (SI). In my opinion it would be better to move some of the figures in the SI to the main text, in particular figures S7 and S13, that show some peculiar and interesting optical and electronic properties of the studied system.

In the text is not explained why the solvothermal reaction was performed up to 180°C. Has the author tried higher temperatures? Since the sulphur content increases with the temperature, does the author expect a higher thionation of the reduced graphene oxide at temperatures above180°C?

I found quite interesting the Mott-VHR/ES-VHR transition exhibited by the studied system, also because, as far as I know, it was not observed for the reduced graphene oxide system. I think that these electrical properties should be stressed out more in the paper, and discussed in more details.

Minor Comments:

  • Title: Langmuir-Blodegett should be Langmuir-Blodgett
  • Introduction, page 1, line 16, “...groups on graphene have been played a role of linking...”, please check the grammar of the sentence.
  • Results and Discussion, page 4, Figure 2 should be expanded.
  • Results and Discussion, page 3, I think figure 1 is not needed.
  • Results and Discussion, page 6, lines 24 and 27, what are the figures 6(a) and 6(b) that you refer to?
  • Results and Discussion, page 6, line 48, “ ...it is to estimate...”, I think it should be “...it is possible to estimate...”

Supporting Info

- Figure S6, the EDS spectrum is not readable.

Reviewer 3 Report

This manuscript presents insights into a new thionation method that lead to the preparation of reduced graphene oxides with interesting conducting properties at different reaction temperatures. I find the results interesting and agree that the findings presented in the manuscript are important as they could lead to future applications and that this method can be used into other carbon nanomaterials.

However, the manuscript that I have received is a poor draft in earlier stages of submission and therefore requires a major revision before even been considered for publication in this journal or in any other journal of matter.

I am mentioning several points that might be corrected, and I also encourage the author for a deep review of the whole manuscript:

  • I have found that manuscript is incomplete as figures 4 to 6 mentioned in the main text cannot be found elsewhere.
  • The referencing to the figures of the supplementary material after Fig. S3 is completely out of order and difficult to follow.
  • The first paragraph in the introduction is vague and some work must be done on it. The author mention that “… production of synthesized graphene or mechanically exfoliated graphene has been a challenging task for the past few years for its utilization in many potential areas”. I would be better to mention a few examples of the limitations and also of the applications. Also, the author says that …. “easy pathway to tune various physical as well as chemical properties of carbon…”. Can also mention some examples of such physical/chemical properties.
  • The last paragraph of the introduction must be replaced by one that explains the contents of the manuscript. The current paragraph must be re-written and to be moved to the beginning of the results, in order to give a context to the results. Also, please make sure to emphasize on mercapto reduced graphene oxides (mRGO) as this term was only mentioned in the abstract and not in the main text.
  • There is a reference missing on the first paragraph of the results and discussions section: “… P4S10 has been utilized for the same purpose in previous reports”.
  • There are two references missing in the second paragraph of the results and discussions section: “Unlike the previous report where a refluxing condition was employed on …” and “…. as reported previously, by reacting readily with pyridine …”.
  • There is a sentence in the last part of the second paragraph of the results and discussions section: “Thionation of alcohols by 1 needs a closer…” What does the author mean with this statement, it is about the ratio of the thiols in the alcohols or what?
  • I don’t agree with the sentence in the third paragraph of the results and discussions section: “ The decrease in the amount of the oxygen-functional groups was observed also for the tRGO series at a similar rate”. First, it is only possible to argue about the increment of S, and secondly the rate of appearance of S is only similar between 150 and 180. Therefore, this sentence must be worked on. In fact, the reduction of the oxygen-functional groups with temperature is an important result that must be extended.
  • The fourth paragraph of the results and discussions section is unclear and must be re-written, also I cannot see that the maximum sulfur content found in mGROS is 7.6 %. Are we treating the thiol (C-SH) and sulfur (C-S) functional groups as the same?
  • Why does the amount in the functional groups in table 1 only add to about 100 % for the GO and not for mRGO or tRGO? Where are the extra missing functional groups?
  • I can see the dispersion in Fig 3, but I would suggest a better resolution image.
  • I don’t agree that mRGO exhibits a good dispersibility in water after centrifugation. Maybe is the low quality of the figure, but I see precipitation in water.
  • In the eight paragraph of the results and discussions section the author makes a bold argument and says that mRGO-180 and tRGO-180 have similar amounts of functional groups, while I can see from table 1 that is not the case.
  • Also, the ID/IG ratio is presented as an important result but it doesn’t fit in the flow of discussion and apparently lacks of explanation for the smaller energy gap.
  • The author also presents an important result” “ … the sheet resistance of the mRGO LB film can be controlled with the amount of sulfur functional group”. However, later in the discussion the large resistance is attributed to an inefficient physical contact among graphene sheets in the LB films. I have the sense that such controversially opposite results must be conciliated and any indication of high conductance must also present indications of low resistance.
  • Fig. S13 shows the semi-logarithmic plots of R vs T^-1 (T^-2 and T^-3). However, by visual comparison I cannot conclude on which mechanism the resistance and temperature are related. I think such a plot must include the parameters of the linear fitting that lead to the conclusions made in the text.

Round 2

Reviewer 1 Report

After reading the answers to the reviewer´s comments, it is a pleasure to accept the manuscript in present form

Reviewer 3 Report

I have revised the corrected manuscript and I have found that the author has made a great effort to improve the manuscript and I have encountered that most of my comments have been targeted and given a good solution. Additionally, I have also found several improvements that enhance the manuscript impact.

In my opinion the manuscript is ready to be accepted.